# Prevalence of Flavescence Dorée Phytoplasma-Infected *Scaphoideus titanus* in Different Vineyard Agroecosystems of Northwestern Italy

**DOI:** 10.3390/insects11050301

**Published:** 2020-05-13

**Authors:** Matteo Ripamonti, Mattia Pegoraro, Marika Rossi, Nicola Bodino, Dylan Beal, Loretta Panero, Cristina Marzachì, Domenico Bosco

**Affiliations:** 1Department of Agriculture, Forest and Food Sciences, University of Torino, 10095 Grugliasco, Italy; matteo.ripamonti@unito.it (M.R.); dylan.beal@berkeley.edu (D.B.); 2Institute for Sustainable Plant Protection, CNR, 10135 Torino, Italy; m.pegoraro@inrim.it (M.P.); marika.rossi@ipsp.cnr.it (M.R.); nicola.bodino@ipsp.cnr.it (N.B.); cristina.marzachi@ipsp.cnr.it (C.M.); 3Council for Agricultural Research and Agricultural Economy Analysis, Viticulture and Enology (CREA-VE), 14100 Asti, Italy; loretta.panero@crea.gov.it

**Keywords:** leafhopper vector, wild *Vitis*, Flavescence dorée epidemiology

## Abstract

Quantitative estimates of vector populations and their infectivity in the wild and in cultivated compartments of agroecosystems have been carried out to elucidate the role of the wild compartment in the epidemiology of Flavescence dorée (FD). Seven sites were selected for the investigations in the Piedmont Region of Italy. They were characterized by a high variety of agricultural and ecological landscape features, and included a vineyard surrounded by wild vegetation. In order to describe abundance and prevalence of FD-infected vectors in the cultivated and wild compartments of the vineyard agroecosystem, adults of *Scaphoideus titanus* were collected by yellow sticky traps inside and outside the vineyard over the period July 10th–September 9th, 2015. They were counted and singly analyzed for the presence of FD phytoplasmas by PCR. Multifactorial correlations among vector population level, prevalence of infected insects inside and outside the vineyards, disease prevalence in cultivated and wild *Vitis* plants, and location of wild *Vitis* plants with respect to the vineyard were analyzed. Abundance of *S. titanus* adults significantly decreased from the end of July onwards, particularly inside the vineyard (average range 22.7 ± 2.5 insects/trap). Percentage of FD-positive *S. titanus* was significantly higher outside the vineyard (up to 48% on average) compared to inside the vineyard (up to 34% on average), and increased during the season in both compartments.

## 1. Introduction

Flavescence dorée of grapevine (FD) is a phytoplasma-associated disease present in several European countries. The disease has a major impact on viticulture because inflorescences and berries dry up, shrivel, and cannot be further processed. Other symptoms include downward leaf rolling with yellowing (in white varieties) or reddening (in red varieties), leaf vein necrosis, lack of lignification, presence of black spots on the new canes, and premature leaf fall [1]; on the most susceptible varieties, stunting or lack of bud break is also observed early in the season [2]. Plants can either be persistently infected over several years and eventually die or recover [3,4]. Phytoplasmas are phloem-obligate, nonculturable pathogens described under the provisional genus “*Candidatus* Phytoplasma” mainly based on 16S rRNA gene phylogeny. By definition, FD phytoplasmas (FDp) are those belonging to the 16SrV-C and -D ribosomal subgroups transmitted by the American grapevine leafhopper, *Scaphoideus titanus* Ball [1,5]. The vector transmits FDp according to a persistent propagative modality; a long latent period, approximately one month, is required for the insect to become infectious [6]. The vector remains infectious for life. *S. titanus* is the main vector of FDp, as it feeds and breeds on *Vitis* species and can transmit FDp following acquisition on either cultivated grapevine inside the vineyard or from infected, naturalized *Vitis* rootstock plants outside the vineyard, thus sustaining both secondary (within vineyard) and primary (from outside the vineyard) infections. Abandoned (or untreated) vineyards and wild *Vitis* rootstock plants in the areas surrounding vineyards are frequently infected and represent a reservoir of both FDp and *S. titanus* [7,8]. American *Vitis* spp. do not show symptoms but are susceptible to FDp [9] and are attractive host plants for *S. titanus*, which coevolved with them in the Nearctic Region. Besides *S. titanus*, other leafhopper and planthopper species have been identified as FDp vectors, among these *Orientus ishidae* [10], *Dictyophara europaea, Allygus* spp. [11], and *Phlogotettix cyclops* [12]. However, these latter species are polyphagous rather than grapevine feeders and are likely to spread phytoplasmas in the wild compartment and, only occasionally, transmit FDp to cultivated grapevines (primary infections). *S. titanus* is regarded as the main vector associated with all the major epidemics of the disease [1]. FD was first identified in the Piedmont Region of Italy in 1998 [2]; soon after its discovery, dramatic epidemics occurred because vector populations were not controlled, and the disease progressed rapidly because of vine-to-vine transmission within the vineyard. Following the enforcement of compulsory control of FD, mandatory uprooting of infected plants, and insecticide applications against the vector, secondary infections were substantially suppressed. However, over the years, a number of new infections took place, namely in the vines close to vineyard borders. These observations suggested that the wild compartment, represented by naturalized *Vitis* and associated *S. titanus*, was the major source of infection. To clarify the epidemiology of FD in the area, genetic tracking of phytoplasmas in the vineyard agroecosystems was carried out [8], and results showed that FD haplotypes identified in cultivated and wild *Vitis*, as well as in *S. titanus* collected inside and outside the vineyards, largely overlap, further proof of the wild compartment role in the FD epidemiology. The comparative analysis of population level and of proportion of infected *S. titanus* in the cultivated and wild compartments of the vineyard agroecosystem is almost unexplored and represents essential information for management of FD. The aim of the work is to fill this knowledge gap by conducting systematic investigations in representative sites of the Piedmont Region, where FD is a major problem for viticulture. Our results substantially improve the understanding of the epidemiology and contribute to the design of rational and effective control programs of FD and its vector.

## 2. Materials and Methods 

### 2.1. Sites, Vineyards, and FD Prevalence 

The same seven sampling sites as described by Rossi and coworkers [8] were selected in an important winegrowing area of the Piedmont Region, northwestern Italy. They were characterized by a high variety of agricultural and ecological landscape features, but all included cultivated *Vitis vinifera* (several cvs, see Table 1) with different prevalence of FD disease, presence of the FD vector *S. titanus*, and potential alternative host plants for the FDp (e.g., abandoned *V. vinifera*, naturalized rootstocks of *V. riparia* and hybrids of different *Vitis* species, and *Clematis vitalba*). The sites were named after the villages closest to them using the following abbreviations: AT, CI, CR, LM, MO, PA, and PC, as previously detailed (Figure 1) [8].

FD prevalence was calculated by visual inspection for FD-specific symptoms, as described in Morone et al. [13]. Prevalence of FD in the vineyards was ranked in four categories, spanning from about 1% to more than 30% (Table 1).

The vineyard in Asti (Figure 2, AT) was a multivarietal experimental plot of 2.6 ha with several red (Albarossa, Barbera, and Syrah) and white (Chardonnay, Cortese, and Incrocio Manzoni) cvs. A forested area bordered the vineyard to the north, and to the south a tree line separated it from a meadow. On the western side, a large abandoned vineyard was present, and to the east a grassy area separated the vineyard from a dense edge of naturalized rootstocks. FD prevalence was in between 5% and 10%. At Cisterna d’Asti (Figure 2, CI), the 0.9 ha vineyard of cv Croatina was characterized by a forested area on the steep south-facing slope to the north of the vineyard, with an abandoned vineyard where wild rootstocks were present. At this site, FD prevalence in the vineyard ranged between 10% and 15%. At Castel Rocchero (Figure 2, CR), the 1.9 ha vineyard consisted of Barbera and Dolcetto cvs and the FD prevalence was about 30%. There were very few trees or wild vegetation around the vineyard as the surrounding area was characterized by intensive viticultural practices. A few naturalized grapevine plants were found and sampled along the roadside to the west and on the top of a mild west-facing slope on the eastern side of the vineyard. The vineyard of cv Nebbiolo at La Morra (Figure 2, LM) was 1.1 ha in size and surrounded on three sides by dense forestation and on the southwestern edge was separated from another vineyard by a narrow windbreak of trees. Around the vineyard edges of La Morra there were a few sparse populations of *C. vitalba* plants and numerous wild rootstocks, from an old abandoned vineyard. At La Morra, less than 1% of the plants showed FD symptoms. At Montà d’Alba (Figure 2, MO), the small vineyard (0.1 ha) of cv Nebbiolo was on the middle of a mild slope bordered by hazelnut orchards to the north and dense forestation bordering the roadway that wrapped around it. Along the western side of this forested edge, wild rootstocks from abandoned vineyards were found. Prevalence of FD at this site was between 10% and 15%. Paderna’s vineyard (Figure 2, PA; 1.5 ha) was planted with cvs Freisa, Merlot, Dolcetto. Forested edges bordered the vineyard to the east and south, grassy plains and herbaceous crops surrounded the vineyard to the north and west. Within these forested edges, there were several abandoned *V. vinifera* and *C. vitalba* plants. No more than 1% of the vines showed FD symptoms. The Portacomaro vineyard, of cvs Barbera, Grignolino, and Ruché, (Figure 2, PC; 1.2 ha) was situated at the top of a steep sloped hill and was bordered by land for livestock production to the north, a narrow forest and civic housing to the west, a dense forest to the east, and the town of Portacomaro to the south. The western and southern edges of the vineyard were surrounded by hazelnut orchards. Within the southwestern forest, substantial populations of wild rootstocks were found. Both *C. vitalba* and wild rootstock plants were also found along the northern tree line that separated the viticultural and livestock production areas. Visual estimates of FD prevalence at this site was about 30%. Out of the seven sampled sites, only Castel Rocchero and Paderna were not subject to conventional chemical control of insect pests of viticulture (that includes two insecticide applications against *S. titanus*, the first against nymphs and the second against adults), and were managed according to guidelines for organic viticulture (based on three applications of pyrethrins in June–July).

### 2.2. Insect Monitoring and Collection

*S. titanus* populations were monitored at each site both inside and outside the cultivated vineyards by means of yellow sticky traps (YSTs), 25 × 40 cm (0.1 m^2^) (Figure 2) during summer 2015. Traps were hung 1.5 m high during July–beginning of September, the best period to collect adults of this species according to its life cycle [14]. They were replaced for three trapping periods: July 10th to 31st, July 31st to August 21st, and August 21st to September 9th, from now on defined as period A, B, and C, respectively. Climatic conditions of these periods are summarized in Appendix A, where minimum, maximum, and average monthly temperatures, as well as rainfall, are reported for three sites close to the investigated ones. Following counting, the *S. titanus* adults were removed from sticky traps with a paintbrush and a drop of vegetal solvent. At Castel Rocchero, due to the absence of wild vegetation around the vineyard, traps were hung inside the vineyard only. At the Portacomaro and Montà sites, due to the very high number of *S. titanus* found outside the vineyard, some adults were also collected by sweep net with the purpose of molecular detection for FDp presence. All insect samples were stored under ethanol in glass vials at −20 °C until nucleic acid extraction.

### 2.3. Nucleic Acid Extraction and FDp Detection

Total nucleic acids were extracted from single leafhoppers according to the method of Pelletier [15], then suspended in 75 µL of Tris-HCl 10 mM pH 8. DNA concentration was measured with NanoDrop 2000 TM Spectrophotometer (Thermo Scientific, Waltham, MA), and all samples were then diluted to 20 ng/µL. The presence of FDp was detected by Real-Time PCR (CFX Connect Real-Time PCR Detection System, Bio-Rad, Hercules, CA, USA) with primers mapFD-F/mapFD-R and the TaqMan probe mapFD-FAM [15]. The PCR mix (10 μL) contained 1× iTaq Universal Probe Supermix (Bio-Rad), together with 300 nM primers, and 200 nM probe, and 20 ng of total nucleic acids. Samples were run in triplicate, together with a negative control, with double-distilled water instead of template nucleic acid. All insects collected at AT, CI, CR, LM, and PA were from sticky traps. About 40 samples from MO and all those from the surrounding abandoned vegetation in PC were collected by sweep net (Appendix A).

### 2.4. Data Analyses

#### 2.4.1. Vineyard Mapping

Schematic maps were produced with the software QGIS v 3.2.3 ‘Bonn’ [16] (Figure 1 and Figure 2).

#### 2.4.2. Statistical Analyses and Graphical Representation

The dataset consists of multiple captures of *S. titanus* through yellow sticky traps hung at fixed places inside or outside each vineyard (Figure 2). *S. titanus* were then pooled for vineyard, time period, and trap position for analyses of FDp status. 

To model the number of *S. titanus* individuals trapped as a function of the covariates, a negative binomial generalized linear mixed model (GLMM) with a log link function was used. Fixed covariates were Trap position (categorical with two levels—“inside” and “outside” the vineyard), and Time period (categorical with three levels). The interaction terms were Trap position × Time period. To incorporate the dependency among observations of the same vineyard, we used Vineyard as a random intercept.

Model assumptions were verified by plotting residuals versus fitted values, for each covariate in the model and for each covariate not in the model. We assessed the residuals for temporal dependency. Model validation did not raise any significant concern with normality of residuals and linear relationship among variables (Table 2, Figure 4).

To model the proportion of FDp-positive *S. titanus* as a function of the covariates, a binomial GLMM with a logit link function was used. The logit link function ensures fitted values among 0 and 1, and the binomial distribution is typically used for proportion data. Fixed covariates were Trap position (categorical with two levels—“inside” and “outside” the vineyard), Time period (categorical with three levels). The interaction terms were Trap position × Time period. To incorporate the dependency among observations of the same vineyard, we used Vineyard as a random intercept. Overdispersion was accounted for by using a quasi-GLM model and correcting the standard errors accordingly (Table 3, Figure 6).

The package lme4 [17] and glmmPQL [18] in the software R [19] were used to fit the models.

Correlation between proportion of infected *S. titanus* and proportion of infected grapevines measured inside the vineyards was estimated using nonparametric Spearman’s rank correlation (*cor.test* in stats R package) [19].

Wilcoxon rank-sum test was applied to the comparison of *S. titanus* numbers trapped inside vs. outside the vineyard at each time period (Figure 3). Z-test was used to compare the proportion of infected *S. titanus* collected in the same compartments of the vineyard agroecosystems (Figure 5). Plots were constructed using package ggplot2 [20] and lemon [21] in the software R.

## 3. Results

*S. titanus* adults were collected in all sites during July and August both inside and outside the vineyards (Appendix A). At Asti, sticky traps collected many more samples outside the vineyard, in the canopy of naturalized rootstocks climbing on broadleaved trees. At Cisterna, a similar population level of *S. titanus* was estimated inside and outside the vineyard, although late in the season more adults were collected outside the vineyard. At Castel Rocchero, all samples were collected inside the vineyard, as no uncultivated areas were present around the investigated vineyard. At La Morra, sticky traps collected more *S. titanus* outside compared to the inside of the vineyard; at this site, the highest vector population was recorded. At Montà, a substantial amount of *S. titanus* adults were collected both inside and outside the vineyard. At Paderna, similar numbers of leafhoppers were trapped inside and outside the vineyard, although they were more abundant outside the vineyard in August and the beginning of September. At Portacomaro, similar numbers of *S. titanus* were collected by YSTs inside and outside the vineyard. To this purpose, it should be mentioned that, in the previous year, the population of *S. titanus* in the wild compartment surrounding this vineyard was much higher, and leafhoppers could be collected directly from the leaves with a mouth aspirator. The population then declined in 2015 as most of the wild vines were uprooted during the winter. In all the vineyards, the highest levels of population were recorded in July, and then decreased rapidly in the following months (Figure 3).

The vector population level decreased from period A to periods B and C with a significantly different rate inside and outside the vineyard. There was a significant interaction between position and sampling time on the number of trapped *S. titanus*. That is, the *S. titanus* captures significantly decreased as the season progressed, especially inside the vineyards. Indeed, the *S. titanus* counts were mostly similar among traps located inside and outside the vineyards in the first period (July), but differed for later sampling periods (August–early September), with traps located outside collecting more insects compared to the ones located inside the vineyards (Table 2 and Figure 4).

The proportion of FDp-infected leafhoppers varied according to the vineyard, the trap position (inside/outside), and the sampling time. More than 40% of infected leafhoppers were recorded at Cisterna, La Morra (outside the vineyard only), Montà, and Portacomaro, while at Paderna, about 10% of leafhoppers (both from inside and outside vineyard traps) were FDp carriers. Similarly, inside the vineyard of La Morra, only 7% of tested leafhoppers were infected (Appendix A). Overall, more leafhoppers collected in the wild compartment were FDp-infected compared to those collected inside the vineyard (Figure 5). Although leafhoppers collected later in the season were more frequently infected, a remarkable proportion of adults collected in July tested positive for FDp (Figure 5 and Figure 6), thus suggesting that many, if not most, leafhoppers acquired phytoplasmas at the nymphal stages.

The proportion of FDp-positive *S. titanus* was significantly higher for individuals trapped outside than inside the vineyards, irrespective of the period of trapping (Table 3 and Figure 6). The proportion of FDp-positive *S. titanus* per vineyard was higher for the second and third time periods, although not significantly (Table 3).

A positive correlation was found between the proportion of FDp-infected leafhoppers collected inside the vineyard and the proportion of infected grapevines in the same vineyards (ρ = 0.75, *p* = 0.051, R^2^ = 0.57; Figure 7).

## 4. Discussion

This study was conducted under field conditions in different viticultural areas of the Piedmont Region of Italy to describe the abundance of vector populations and to estimate vector infectivity inside the vineyards and in the wild compartments surrounding the vineyards. 

The selected sites were characterized by the presence of FD-infected cultivated *V. vinifera*, and of a wild compartment with potential alternative host plants for the FDp and its vector. For these reasons, vector population levels and proportion of FDp-infected leafhopper do not reflect the average situation in the Piedmont Region, but rather the worst-case scenarios. We confirmed that naturalized *Vitis* may host very high populations of *S. titanus* and that vineyards close to wild vegetation (AT, CI, LM, MO, PA, PC) or not properly treated with insecticides (CR) may also host high populations of *S. titanus* adults. 

The highest numbers of *S. titanus* adults were collected with YSTs in July (likely at the very end of July), and then captures declined more or less gradually, both within the vineyards and in the wild vegetation compartment. In most reports, *S. titanus* populations peak in the first half of August [14,22,23]. The very warm conditions of 2015 that anticipated *S. titanus* development (Appendix A) and the application of insecticides in the vineyards at the end of July may explain the abundance pattern recorded in this study. In Romania, the *S. titanus* adult population peaked at the end of July in the years 2009–2011 in an abandoned vineyard close to Bucharest [24]. However, our data cannot be used to properly identify a population peak since YSTs were exposed in the fields for 20-day periods under our experimental conditions. The pattern of population decrease over August and the beginning of September was significantly different inside and outside the vineyard (population decreased faster inside the vineyard). The faster population decrease inside the vineyards was likely due to the insecticides applied against the adults at the end of July. Very high numbers of adults were collected from the wild vegetation compartments, where insecticide applications are forbidden by law and the only available control measure is the difficult mechanical roguing of wild *Vitis*. Based on our experience, the presence of *S. titanus* within an abandoned/wild area is highly aggregated, and therefore YST captures highly depended on their specific location within the wild vegetation area. This means that our estimate of the *S. titanus* population levels outside the vineyards suffered from some inaccuracy. Nevertheless, as very high captures were repeatedly obtained, together with the observations reported for Italian and North American vineyard agroecosystems [7,25,26], we can conclude that wild vegetation areas are very important sources of the vector for the nearby agroecosystems. So far, we have no hints to explain uneven aggregated spatial distribution of *S. titanus* in the wild compartment. This issue is very difficult to study, as wild compartments are very different in size, slope, orientation, and plant composition. However, the presence of large surfaces of wild *Vitis* climbing on high broadleaved trees, as was the case for all the analyzed sites except CR, rather than covering the soil, is a factor that favors the presence of high *S. titanus* populations (personal observation).

Among the vineyards, the highest population levels were recorded at LM and CR. In the latter vineyard, only pyrethrins were applied against *S. titanus* and this can account for the high population of the insect. As for the LM vineyards, no specific factors (size of the vineyard and of wild vegetation area, slope, exposure) could be evoked to account for this high *S. titanus* density which is, to some extent, unpredictable. 

Our GLMM model showed that a higher proportion of infected insects was recorded for the leafhoppers collected in the wild compartment compared to those from within the vineyard. This evidence is consistent with data of Lessio et al. [25], confirming the major role of the wild vegetation in the spread of FD. Similarly, untreated vineyards are a known source of infected *S. titanus* [7]. However, at some of the sites, the proportion of FDp-carrier insects was similar in the two compartments. If we assume that the proportion of FDp-infected leafhoppers can be used as a marker of insect dispersal, we can speculate that at most of our sites (AT, CI, MO, PA, and PC), there was a flow of *S. titanus* between the cultivated and wild compartments of the vineyard agroecosystem. On the contrary, at LM, the two populations were apparently separated. In fact, very few insects were FDp carriers inside the vineyards, and many were infected in the wild compartment. Interestingly, at LM, the wild vegetation was present in a large area standing downhill and below the level of the vineyard, with a woodland shield protecting the vineyard from major air flows. It is then possible that leafhoppers, in the absence of ascendant air flows, are unable to fly upwards and reach the vineyard. Indeed, without prevalent wind conditions, *S. titanus* does not move far [27]. Where the wild vegetation surrounding the vineyard is at the same height or above the vineyard, leafhoppers may move freely between the two compartments or into the vineyard itself. Also, where the wild vegetation is below the level of the vineyard but upward and downward air currents are present, the leafhoppers might circulate between the two compartments. The role of wind in *S. titanus* dispersal has been noticed and considered in pest risk assessment of FD [28]. 

The proportion of FDp-infected vectors slightly increased over the summer, in line with the data of Lessio et al. [22]; it is worth remembering that this proportion increased in spite of the higher mortality of FD-infected *S. titanus*, demonstrated by Bressan et al. [29]. Increase in the proportion of infected insect vectors over the season is expected, since FDp circulates, multiplies, and thus persists for life in the insect body. With time, chances for the insects to move and feed on an infected source plants obviously increase, and this also contributes to increasing the proportion of FDp-positive insects during the summer season. However, in July, the proportion of FDp-positive adults was already high, and this may have two concurrent explanations: (i) most of the insects acquired phytoplasmas at the nymphal stages and were already infected when the adult emerged, and (ii) a number of insects collected inside the vineyard in July already came from outside the vineyard, where chances of feeding on an infected wild *Vitis* were greater. Actually, Table 1 shows that, overall, one fourth of the wild, asymptomatic *Vitis* tested at random were FDp-infected, while percentages of infected plants within the vineyards were generally lower. Therefore, the increase in FDp-positive insects may be partly explained by the increasing load of FDp in the insects due to multiplication over time. This multiplication of FDp in most insects that fed on infected source plants would overcome the detection threshold of the PCR assay. The chosen real-time PCR assay detects phytoplasmas well before the completion of their latent period (about one month, during which the vector is infected but not infectious yet). However, FDp could also be acquired by adults [30,31], and this is also consistent with our observations. In previous papers, we demonstrated that infected vines have low FDp load early in the season [32] and that acquisition of FDp by *S. titanus* correlates with phytoplasma load in the plant [33]. We can then speculate that, as the season progresses, the likelihood of FDp acquisition by the vectors increases and this may also account for the increasing proportion of infected insects recorded during our survey. The proportion of FDp-carrier leafhoppers inside the vineyard showed a positive correlation with the proportion of infected vines at the same site, confirming that PCR detection of FDp in the vector is a good marker of disease spread/prevalence in the vineyard. However, since the R^2^ of the model was equal to 0.57, presumably other factors, besides the proportion of infected leafhoppers inside the vineyard, may account for the spread of the disease within a vineyard (e.g., the susceptibility of grapevine cultivars). Actually, the vineyards were cultivated with different varieties, and these may show different levels of susceptibility to FD [9]; differential susceptibility was not taken into account in this work because only empirical observations are available for local varieties cultivated in the Piedmont Region so far. Analyses of a robust set of experimental data on the susceptibility of different vine varieties are ongoing in our laboratory.

## 5. Conclusions

High numbers of *S. titanus* adults were collected from the wild vegetation compartment of several sites, and vector population levels of this compartment were higher than those measured inside the corresponding vineyard. The pattern of vector population decrease over August and the beginning of September was significantly different inside and outside the vineyard (population decreased faster inside the vineyard), thus confirming the effects of the insecticides applied against the adults in the vineyards. As expected, the proportion of FDp-infected vectors increased over the summer, even though the proportion of FDp-positive leafhoppers (possibly not infectious yet) was already high in July, indicating that grapevines are exposed to infectious leafhoppers for a long period of time. A higher proportion of FDp-infected leafhoppers was recorded for the insects collected in the wild compartment compared to those from the vineyard, thus indicating the important role of outside FDp sources in the epidemiology of the disease. This study provides valuable information on the role of the wild compartment in the epidemiology of Flavescence dorée disease, and represents one of the few studies conducted at the level of the vineyard agroecosystem as a whole. Further research should be devoted to the evaluation of FD spread reduction following removal of wild *Vitis* in the surroundings of vineyards.

## Figures and Tables

**Figure 1 insects-11-00301-f001:**
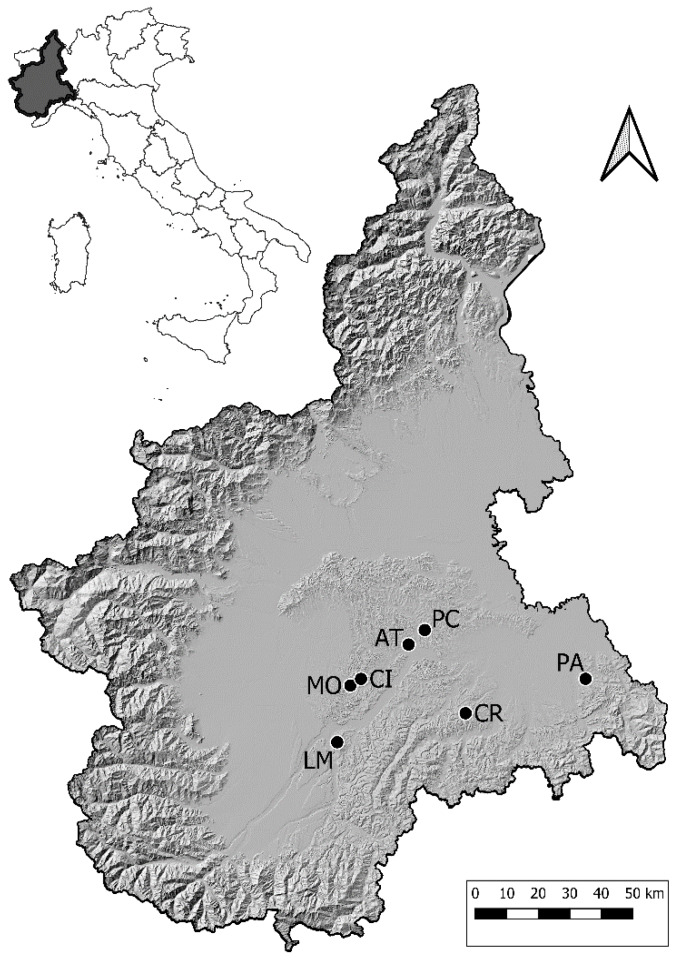
Location of the seven experimental sites within the Piedmont Region of Italy. AT, Asti; CI, Cisterna d’Asti; CR, Castel Rocchero; LM, La Morra; MO, Montà d’Alba; PA, Paderna; PC, Portacomaro.

**Figure 2 insects-11-00301-f002:**
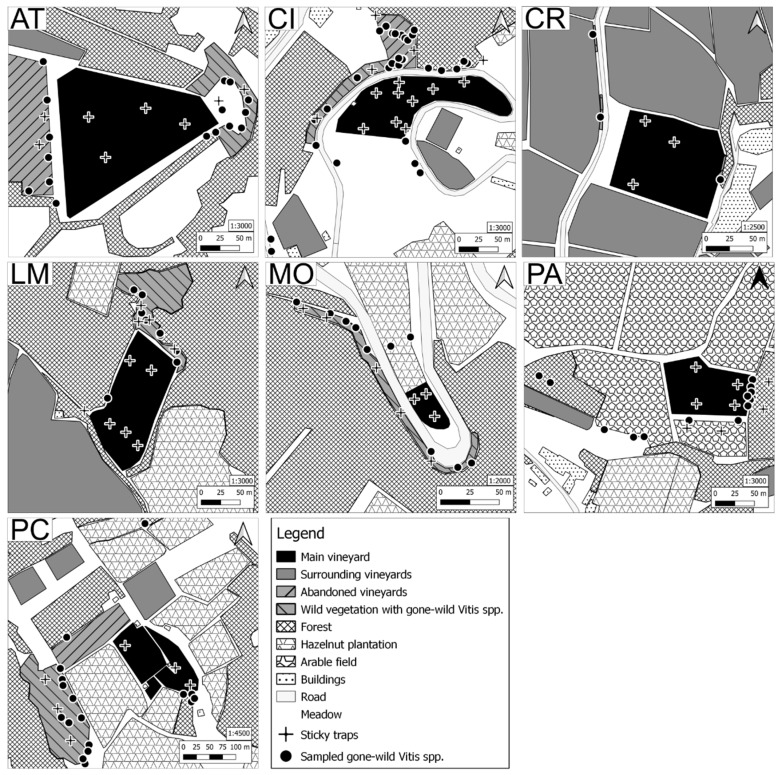
Stylized vegetal composition of the seven vineyards and surroundings. Acronyms: AT, Asti; CI, Cisterna d’Asti; CR, Castel Rocchero; LM, La Morra; MO, Montà d’Alba; PA, Paderna; PC, Portacomaro.

**Figure 3 insects-11-00301-f003:**
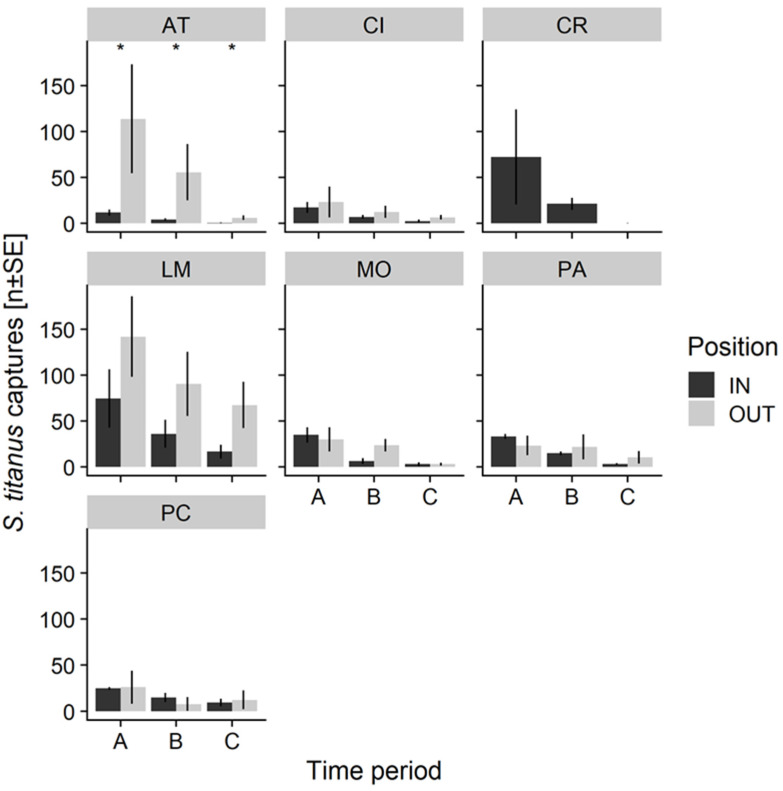
Number of *S. titanus* adults trapped in the three summer periods ± Standard Error (A = July 10th–July 30th; B = July 31st–August 20th; C = August 21st–September 10th) over the different sampling sites. Inside (IN) and outside (OUT) captures in the same vineyard are represented (Appendix A). Asterisk indicates significant difference between the number of *S. titanus* collected inside and outside the vineyard (*p* < 0.05).

**Figure 4 insects-11-00301-f004:**
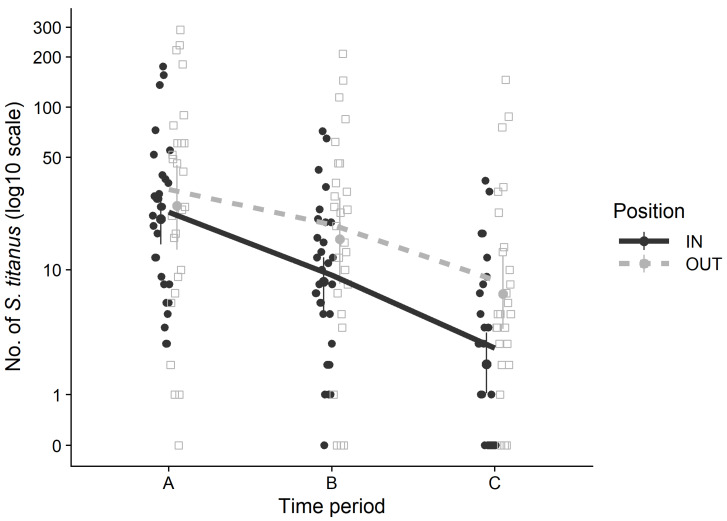
Number of *S. titanus* collected throughout the season in Piedmont vineyards. Black points and continuous line represent data and GLMM model of traps located inside the vineyard, whereas grey squares and grey dashed line represent data and model of traps located outside the vineyard.

**Figure 5 insects-11-00301-f005:**
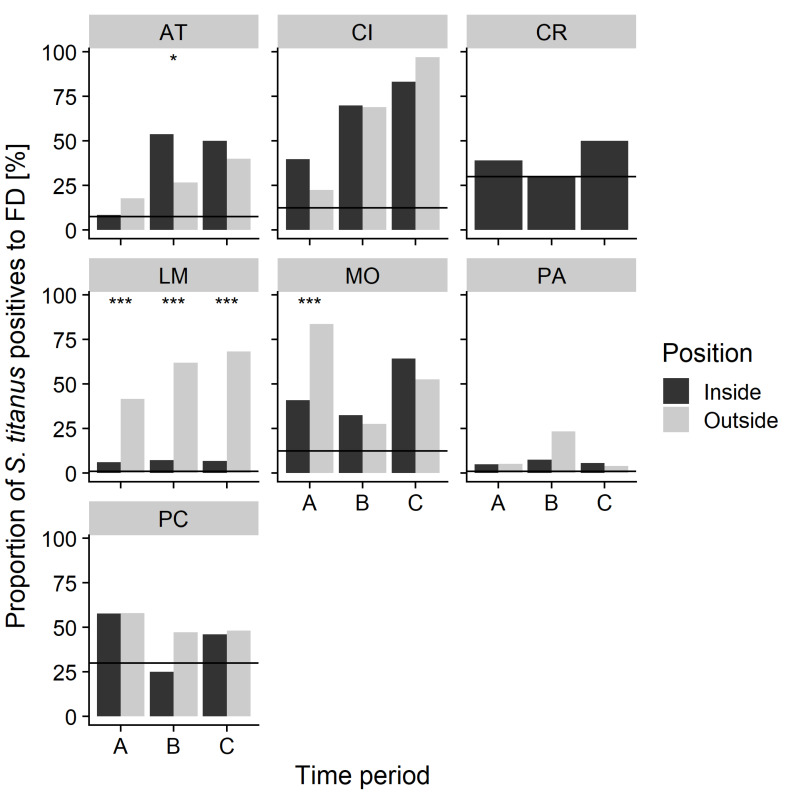
Flavescence dorée phytoplasma infection of *S. titanus* populations sampled with yellow sticky traps exposed for two weeks inside and outside the different vineyards. Inside and outside levels of infection in the same vineyard are represented. The horizontal black line represents the prevalence of FD-symptomatic grapevines in the vineyard (Table 1). The total number of FDp-tested *S. titanus* at each vineyard, time point, and trap position is reported in Appendix A. Asterisks indicate significant difference between the proportion of FD-infected *S. titanus* collected inside and outside the vineyard (* *p* < 0.05; *** *p* < 0.001).

**Figure 6 insects-11-00301-f006:**
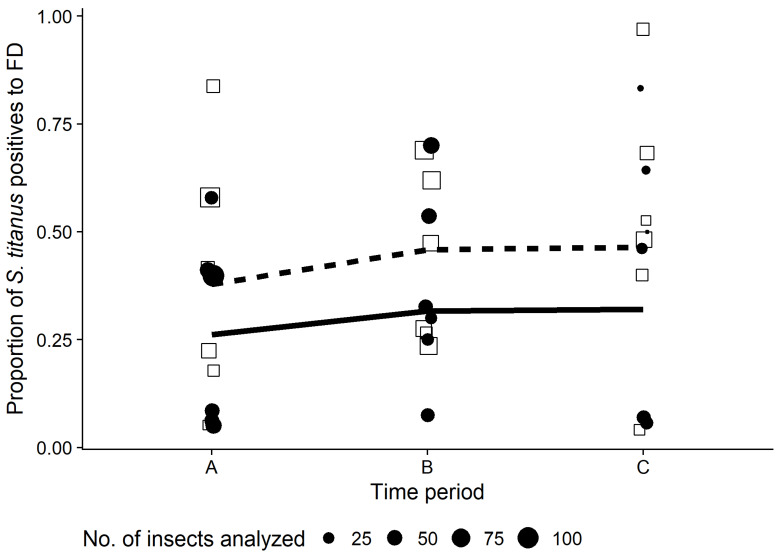
Proportion of FDp-positive *S. titanus* throughout the season in Piedmont vineyards. Black points and continuous line represent data and GLMM model of traps located inside the vineyard, whereas white squares and dashed line represent data and model of traps located outside the vineyard. Size of points and squares is proportional to number of insects analyzed for each trap.

**Figure 7 insects-11-00301-f007:**
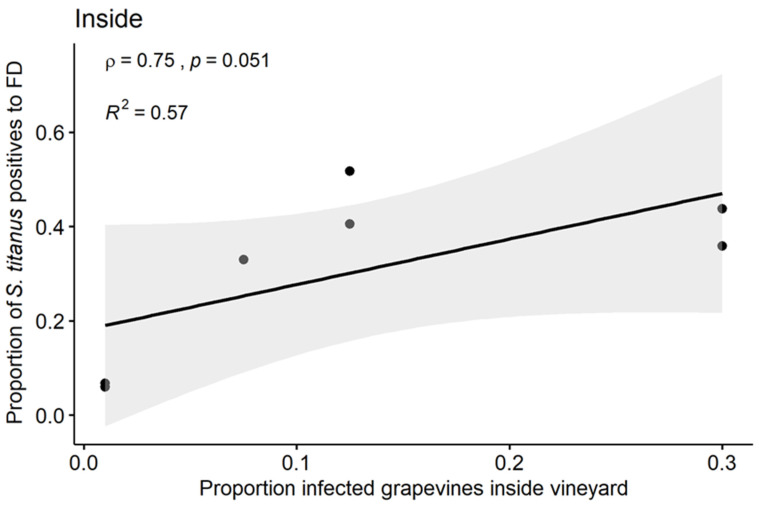
Correlation between proportion of FDp-positive *S. titanus* (collected inside the vineyard) and proportion of infected grapevines in the same vineyard.

**Table 1 insects-11-00301-t001:** Main characteristics of the seven vineyards.

Site	Surface (ha)	Grapevine Cultivars	Inside Traps (n)	Outside Traps (n)	Naturalized *Vitis* (FDp Pos/Tested) ^1^	FD-Infected Grapevines (%)
AT	2.6	Albarossa, Barbera, Chardonnay, Cortese, Incrocio Manzoni, Syrah	4	4	6/19	5–10
CI	0.9	Croatina	9	5	7/43	10–15
CR	1.9	Barbera, Dolcetto	3		0/19	>30
LM	1.1	Nebbiolo	5	5	1/20	≤1
MO	0.1	Nebbiolo	3	5	7/21	10–15
PA	1.5	Freisa, Merlot, Dolcetto	4	4	8/28	≤1
PC	1.2	Barbera, Grignolino, Ruché	3	3	9/39	>30

^1^ Data extracted from [8].

**Table 2 insects-11-00301-t002:** Estimated regression parameters (log), standard errors, z-values, and *p*-values for the negative binomial GLMM of *S. titanus* counts with covariates *Trap position* and *Time period* and their interaction. The lasts two columns define lower and upper limits for 95% confidence interval. The estimated value for σVineyard is 0.334 and σTrap is 1.007.

Effects	Estimate	Std.Error	Statistic	*p*-Value	Conf.Low	Conf.High
Intercept	3.143 ^***^	0.262	11.981	>0.001	2.629	3.657
Position Outside	0.32	0.338	0.948	0.343	−0.342	0.982
Time_periodB	−0.923 ^***^	0.196	−4.719	>0.001	−1.307	−0.54
Time_periodC	−2.118 ^***^	0.22	−9.638	>0.001	−2.549	−1.688
Position Outside:Time_periodB	0.422 ^*^	0.289	1.461	0.144	−0.144	0.988
Position Outside:Time_periodC	0.801 ^**^	0.312	2.57	0.01	0.19	1.412

* *p* < 0.05; ** *p* < 0.01; *** *p* < 0.001.

**Table 3 insects-11-00301-t003:** Estimated regression parameters (odds ratio), standard errors, t-values, and *p*-values for the binomial GLMM of proportion of FDp-positive *S. titanus* with covariates *Trap position* and *Time period* and their interaction. The estimated value for σVineyard is 0.412.

Effects	Estimate	Std. Error	t-Value	*p*-Value
Intercept	−1.341	0.229	−5.85	<0.001
PositionOutside	0.370	0.173	2.14	0.041
Time_periodB	0.191	0.188	1.02	0.317
Time_periodC	0.203	0.227	0.896	0.378

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
