# Peer review of "Prevalence of Flavescence Dorée Phytoplasma-Infected Scaphoideus titanus in Different Vineyard Agroecosystems of Northwestern Italy"

_insects, 2020, doi:10.3390/insects11050301_

Round 1

Reviewer 1 Report

The article is clear and well written. The title corresponds to the text. Introduction is clear and complete, results are well exposed and discussion is well built up. References are adequate.

This paper is of great interest for people dealing with the epidemiology of FDp and the role of the vineyards environment as source of FDp-infected Scaphoideus titanus.

The two only small points I see are

- Lines 249-251: The authors said there is a positive correlation found between the proportion of FDp-infected leafhoppers collected inside the vineyard and the proportion of infected grapevines in the same vineyards. This is correct, however we can also see in the text and in Figure 7 that the R2 is only of 0.57. So the correlation exists, however we have to be aware that other elements are involved in this relation.

- Lines 304-305: I don’t know if I calculate well with table S1, but I found a higher proportion of infected insects in leafhoppers collected in the vineyards compared to those from the wild compartment for 7 cases out of 18 (so almost 40% of the cases). So I do not totally agree with the sentence of lines 304-305 which says that “a higher proportion of infected insects was recorded for the leafhoppers collected in the wild compartment compared to those from vineyard”. This is true but for 60% of the case. Perhaps could you add “mostly”?

Remarks:

Line 49: Orientus , not Orienthus

Line 152: can you provide more precise numbers or proportion? (1%, 10%?).

Figure 3: wouldn’t it be interesting to have the significance between inside and outside (for each location and each period). You can put * for example.

Line 285: a bracket ) is forgotten.

Author Response

The article is clear and well written. The title corresponds to the text. Introduction is clear and complete, results are well exposed and discussion is well built up. References are adequate.

This paper is of great interest for people dealing with the epidemiology of FDp and the role of the vineyards environment as source of FDp-infected Scaphoideus titanus.

We thank the reviewer for his/her appreciation of the work

The two only small points I see are

- Lines 249-251: The authors said there is a positive correlation found between the proportion of FDp-infected leafhoppers collected inside the vineyard and the proportion of infected grapevines in the same vineyards. This is correct, however we can also see in the text and in Figure 7 that the R2 is only of 0.57. So the correlation exists, however we have to be aware that other elements are involved in this relation.

We addressed the point raised by the reviewer in the discussion section “However, since the R2 of the model was equal to 0.57, presumably other factors, besides the proportion of infected leafhoppers may account for the spread of the disease within a vineyard, e.g. the susceptibility of grapevine cultivars.”

- Lines 304-305: I don’t know if I calculate well with table S1, but I found a higher proportion of infected insects in leafhoppers collected in the vineyards compared to those from the wild compartment for 7 cases out of 18 (so almost 40% of the cases). So I do not totally agree with the sentence of lines 304-305 which says that “a higher proportion of infected insects was recorded for the leafhoppers collected in the wild compartment compared to those from vineyard”. This is true but for 60% of the case. Perhaps could you add “mostly”?

The higher proportion of infected insects in leafhoppers from the vineyards compared to those from the wild compartment was calculated by our GLMM model on overall data, rather than considering the single cases. For this reason, to comply with the reviewer observation, we changed the sentence into:

Our GLMM model showed that a higher proportion of infected insects was recorded for the leafhoppers collected in the wild compartment compared to those from within the vineyard.”

Moreover, two lines below, there is another sentence “However, at some of the sites, the proportion of FDp-carrier insects was similar in the two compartments.” that smooths the previous statement and, to some extent, is in line with the referee comment.

Remarks:

Line 49: Orientus , not Orienthus

The mistake has been corrected

Line 152: can you provide more precise numbers or proportion? (1%, 10%?).

We included the detailed information in mat&met . “All insects collected at AT, CI, CR, LM, and PA were from sticky traps. About 40 samples from MO and all those from PC were collected by sweep net (S2).”

Figure 3: wouldn’t it be interesting to have the significance between inside and outside (for each location and each period). You can put * for example.

We accepted the suggestion and we included the significance between data from inside and outside the vineyard in figures 3 and 5 and we also added details of the methods used for the comparisons in mat&met: “Wilcoxon rank-sum test was applied to the comparison of S. titanus numbers trapped inside vs outside the vineyard at each time period (Figure 3). Z- test was used to compare the proportion of infected S. titanus collected in the same compartments of the vineyard agroecosystems (Figure 5). Plots were constructed using package ggplot2 [19] and lemon [20] in the software R.”

Line 285: a bracket ) is forgotten.

The bracket has been introduced

Reviewer 2 Report

This paper provides important new data on the prevalence and seasonality of the main vector of FD disease in vineyards and adjacent areas of natural vegetation in the Piedmont of northern Italy. Overall the study is well done and the paper is well organized and written. Just a few points need to be addressed.

  1. The abstract needs an introductory sentence explaining the purpose of the study.
  2. Line 49: the genus name Orientus is misspelled.
  3. Two points on p. 12 would benefit from further elaboration: line 289: “Based on our experience, the presence of titanus within an abandoned/wild area is highly aggregated.” Can you provide any more specifics? Where do these wild populations tend to aggregate? Line308 “If we assume that the proportion of FDp-infected leafhoppers can be used as a marker of insect dispersal…” Why would this assumption be valid?
  4. Some previous studies indicate that phytoplasma infections (in general, not just FDp) alter both dispersal behavior of vectors and attractiveness of host plants to potential vectors. So, it may be helpful for the authors to discuss the possible implications of these phytoplasma-induced changes on their sampling results. For example, is there any evidence that the attractiveness of YSTs can vary depending on whether or not an individual S. titanus is infected with FDp? Given that many of the FDp-infected plants outside the vineyard were found to be infected but asymptomatic, is it possible that these plants were less attractive to dispersing S. titanus and this accounts for the larger number of individuals attracted instead to the YSTs?
  5. It would also be interesting to know whether the proportion of infected S. titanus differed between the samples obtained using YSTs compared to sweeping. Is there any evidence that YSTs are more (or less) attractive to infected versus uninfected S. titanus? Sweep sampling was performed only at 2 sites (PC and MO). Are the rtPCR results from these sites comparable to those from the other sites?

Author Response

This paper provides important new data on the prevalence and seasonality of the main vector of FD disease in vineyards and adjacent areas of natural vegetation in the Piedmont of northern Italy. Overall the study is well done and the paper is well organized and written. Just a few points need to be addressed.

We thank the reviewer for his/her appreciation of the work

The abstract needs an introductory sentence explaining the purpose of the study.

The sentence has been introduced. “Quantitative estimates of vector populations and of their infectivity in the wild and cultivated compartments of agroecosystems have been carried out to elucidate the role of the wild compartment in the epidemiology of Flavescence dorée.”

Line 49: the genus name Orientus is misspelled.

The mistake has been corrected

Two points on p. 12 would benefit from further elaboration: line 289: “Based on our experience, the presence of titanus within an abandoned/wild area is highly aggregated.” Can you provide any more specifics? Where do these wild populations tend to aggregate?

We have no hints on spatial distribution of S. titanus in the wild compartment. This issue is very difficult to study, as wild compartments are very different among them in size, slope, orientation, plant composition. The only empirical but, in our opinion, robust observation related to this aggregation is the strong preference of S. titanus for wild Vitis climbing on high broadleaved trees, rather than covering the soil. However, we now discuss this issue in the discussion section:

So far, we have no hints to explain uneven aggregated spatial distribution of S. titanus in the wild compartment. This issue is very difficult to study, as wild compartments are very different among them in size, slope, orientation, plant composition. However, the presence of large surfaces of wild Vitis climbing on high broadleaved trees, as was the case for all the analyzed sites except CR, rather than covering the soil, is a factor that favors the presence of high S. titanus populations (personal observation).

The latter sentence (“However….) was already stated in the text but has been anticipated to contribute to the discussion of this point.

Line308 “If we assume that the proportion of FDp-infected leafhoppers can be used as a marker of insect dispersal…” Why would this assumption be valid?

Some previous studies indicate that phytoplasma infections (in general, not just FDp) alter both dispersal behavior of vectors and attractiveness of host plants to potential vectors. So, it may be helpful for the authors to discuss the possible implications of these phytoplasma-induced changes on their sampling results. For example, is there any evidence that the attractiveness of YSTs can vary depending on whether or not an individual S. titanus is infected with FDp? Given that many of the FDp-infected plants outside the vineyard were found to be infected but asymptomatic, is it possible that these plants were less attractive to dispersing S. titanus and this accounts for the larger number of individuals attracted instead to the YSTs?

It would also be interesting to know whether the proportion of infected S. titanus differed between the samples obtained using YSTs compared to sweeping. Is there any evidence that YSTs are more (or less) attractive to infected versus uninfected S. titanus? Sweep sampling was performed only at 2 sites (PC and MO). Are the rtPCR results from these sites comparable to those from the other sites?

I agree with the reviewer in that very complicated and intriguing three-way-interactions (plant/insect/phytoplasma) may occur and regulate, among other aspects, host-plant choice by the vector (as an example, see Orlovskis Z and Hogenhout SA, 2016. A Bacterial Parasite Effector Mediates Insect Vector Attraction in Host Plants Independently of Developmental Changes. Front. Plant Sci. 7:885). However, these aspects have been investigated only in easy to manipulate “model systems”, such as aster yellows and Macrosteles leafhoppers. The knowledge of these aspects largely depend on the availability of the whole phytoplasma genome, in order to be able to conduct reverse genetics experiments, which is not the case for FD. Moreover, the possible different attractivity of YST for phytoplasma-infected and phytoplasma-free leafhoppers is unexplored and therefore would be purely speculative. A leafhopper vector has been found more attracted by AY-infected grapevine branches (Kruger et al., 2015, Phytopathogenic Mollicutes) and the authors speculated that this attractiveness was due to the yellow-like color of the infected leaves. In the same paper they demonstrated that YST were more attractive to the leafhopper compared to other colors, but this was in general for leafhoppers (infection status was not tested).

So, breaking the problem in two:

  1. a) yellow color of the plant (or of the traps) may attract leafhoppers, but independently from their infection status (moreover, in our investigations, most of the vineyards were of red varieties, that respond to FD with reddening and not with yellowing): we can conclude that there are no hints suggesting that YST are more or less attractive to infected vs healthy leafhoppers
  2. b) infected vines might be more or less attractive to leafhoppers. This is a totally unexplored issue for FD. If we refer, as a speculation, to the Orlovskis and Hogenhout paper cited above, infected vines could be differentially attractive also indipendently from the symptoms (due to a putative effector such as SAP-54 of AY). In our sites, cultivated infected vines showed symptoms and wild infected ones were asymptomatic. Assuming that, under these two situations, in which a minority of plants within the vineyard or the wild compartment were infected, the hypothesis that leafhoppers were more or less attracted to YST vs host-plants, is highly speculative and we prefer not to discuss this point in our paper

Actually the only known difference between infected and uninfected S. titanus is related to the fitness, as FD-infected leafhoppers have a lower longevity and prolificity (Bressan, A., Girolami, V., & Boudon‐Padieu, E., 2005. Reduced fitness of the leafhopper vector Scaphoideus titanus exposed to Flavescence dorée phytoplasma. Entomologia experimentalis et applicata, 115(2), 283-290).

For this latter reason we added, in the discussion section, the following sentence: “Proportion of FDp-infected vectors slightly increased over the summer, in line with the data of Lessio et al. [22]; it is worth remembering that this proportion increased in spite of the higher mortality of FD-infected S. titanus, demonstrated by Bressan et al. (29).” 

For example, is there any evidence that the attractiveness of YSTs can vary depending on whether or not an individual S. titanus is infected with FDp? Is there any evidence that YSTs are more (or less) attractive to infected versus uninfected S. titanus?

No, to my knowledge there is no evidence.

Given that many of the FDp-infected plants outside the vineyard were found to be infected but asymptomatic, is it possible that these plants were less attractive to dispersing S. titanus and this accounts for the larger number of individuals attracted instead to the YSTs?

S. titanus, being of north-american origin, is more attracted by American Vitis (the ones in the wild) (see Chuce and Thiéry, 2014; Lessio et al., 2007) rather than from Vitis vinifera, therefore this point, in our opinion, does not apply to our surveys

Line308 “If we assume that the proportion of FDp-infected leafhoppers can be used as a marker of insect dispersal…” Why would this assumption be valid?

In the absence of proven factors that bias the proportion of infected insects in our surveys, we believe that the proportion of FDp-infected leafhoppers can be used as a marker of insect dispersal. A similar proportion suggests that there is a flow of insects between inside and outside populations, a different proportion suggests that there is no flow between inside and outside populations. Of course we are not 100% sure of this and actually we are quite prudent in our manuscript “IF we assume that the proportion of FDp-infected leafhoppers can be used as a marker of insect dispersal,  WE CAN SPECULATE that….

Sweep sampling was performed only at 2 sites (PC and MO). Are the rtPCR results from these sites comparable to those from the other sites?

Given that there are no evidences on the selective attractivity of YST for infected/uninfected leafhoppers, the PCR results from these sites are comparable to those from the other sites. Moreover,  we can ensure that no technical differences exist in detecting phytoplasmas between leafhopper collected with YST or sweeping net. Obviously, samples from sweep net are in better conditions (alive) while those from YST are dead since a variable number of days. However, in our experience, FD detection in samples collected with YST works very well, provided that samples glued to YST are not left in the field for several weeks. We immediately removed samples from the YST at the end of the 20day period and we did not experience any difference in detection efficiency, also because PCR is highly sensitive and able to detect low amount of even degraded phytoplasma DNA. In conclusion we can state that, for a simple detection, as was in our scope, it is not relevant.

Reviewer 3 Report

This paper reports of "Prevalence of Flavescence dorée phytoplasma infected Scaphoideus titanus in different vineyard agroecosystems of Northwestern Italy". This is a carefully done study and the findings are of considerable interest but a more precise direction of future research should add more value.

Author Response

Comments and Suggestions for Authors

This paper reports of "Prevalence of Flavescence dorée phytoplasma infected Scaphoideus titanus in different vineyard agroecosystems of Northwestern Italy". This is a carefully done study and the findings are of considerable interest but a more precise direction of future research should add more value. 

We thank the reviewer for his/her appreciation of the work and we added the following sentence in the conclusions:

Further research should be devoted to the evaluation of FD spread reduction following removal of wild Vitis in the surroundings of vineyards.”

Reviewer 4 Report

Reviewer’s comments

The study investigated the population level and the proportion of infected S. titanus in the vineyard and wild compartments of selected Piedmontese vineyard agroecosystems. The study is relevant and builds on previous studies in the area. However, the authors did not adequately address the rationale of the study. They somewhat failed to identify the specific value or contribution of the study in relation to previous studies. It is suggested that the authors clearly addressed this in introduction and discussion. Another important flaw in the paper is the lack of climatic data, including temperature humidity and rainfall. These are important mediators in any ecosystem. However, I think the authors gathered useful data that should be published after major revision of the manuscript.

Specific comments:

Abstract:

It is difficult to understand the abstract because the objectives of the study are not stated clearly. Secondly, complete acronyms must be accompanied by full definition at first mention.

Introduction:

Line 40 intro: persistent propagative modality break...please simplify this phrase

Materials and Methods:

Lines 90-125: In the materials and methods section, it is best to present the information on the characteristics of the study sites (lines 90-125) in a table. This will make it easier to associate the results to the sites’ attributes.

Lines 132-134: Provide reason(s) for choosing the three sampling periods. The weather data, including humidity, temperature and wind, etc. of the study sites are useful for interpretation of results.  

Results:

Lines 206-207: Please re-write sentence. It is difficult to make sense of this sentence, “Overall, the pattern of…”

Discussion:

Lines 256-273: These sentences adds little value to the discussion; hence should be removed from the manuscript. The authors should focus on their results and discuss the results. It appears the authors are desperately trying to situate their work in relation to existing information; this weakens the contribution of the paper. Lines 274 and 282 show the same devaluing views by the authors of their own work. This discussion needs major improvement.

Round 2

Reviewer 4 Report

The authors addressed the issues that were raised on the first version of the manuscript. The manuscript was substantially revised and has improved significantly. I, therefore, recommend that the manuscript should be published.

There two minors corrections: 

Line 140: “…sites closed to the investigated ones” add “to”

Tables S1 and S2: Decimal place(s) should be consistent. 

Author Response

Thanks.